# Altered Lipid Moieties and Carbonyls in a Wistar Rat Dietary Model of Subclinical Fatty Liver: Potential Sex-Specific Biomarkers of Early Fatty Liver Disease?

**DOI:** 10.3390/antiox12101808

**Published:** 2023-09-28

**Authors:** María Martín-Grau, Mercedes Pardo-Tendero, Pilar Casanova, Mar Dromant, Vannina G. Marrachelli, Jose Manuel Morales, Consuelo Borrás, Serena Pisoni, Sabrina Maestrini, Anna M. Di Blasio, Daniel Monleon

**Affiliations:** 1Department of Pathology, University of Valencia, 46010 Valencia, Spain; maria.martin-grau@uv.es (M.M.-G.); m.mercedes.pardo@uv.es (M.P.-T.); pilar.casanova@uv.es (P.C.); j.manuel.morales@uv.es (J.M.M.); 2University Clinical Hospital of Valencia Research Foundation (INCLIVA), 46010 Valencia, Spain; mar.dromant@uv.es (M.D.); vannina.gonzalez@uv.es (V.G.M.); consuelo.borras@uv.es (C.B.); 3Department of Physiology, University of Valencia, 46010 Valencia, Spain; serena.pisoni@uv.es; 4Laboratory of Molecular Genetics, Istituto Auxologico Italiano IRCCS, 20145 Milano, Italy; s.maestrini@auxologico.it (S.M.); a.diblasio@auxologico.it (A.M.D.B.); 5CIBERFES_ISCIII, 46010 Valencia, Spain

**Keywords:** NAFLD, lipid moieties, carbonyls, sex dimorphism, HRMAS, Wistar rats

## Abstract

Non-alcoholic fatty liver disease (NAFLD) is a condition in which excess fat builds up in the liver. To date, there is a lack of knowledge about the subtype of lipid structures affected in the early stages of NAFLD. The aim of this study was to analyze serum and liver lipid moieties, specifically unsaturations and carbonyls, by nuclear magnetic resonance (NMR) in a subclinical Wistar rat model of NAFLD for detecting early alterations and potential sex dimorphisms. Twelve weeks of a high-fat diet (HFD) induced fat accumulation in the liver to a similar extent in male and female Wistar rats. In addition to total liver fat accumulation, Wistar rats showed a shift in lipid subtype composition. HFD rats displayed increased lipid carbonyls in both liver and serum, and decreased in unsaturated fatty acids (UFAs) and polyunsaturated fatty acids (PUFAs), with a much stronger effect in male than female animals. Our results revealed that the change in fat was not only quantitative but also qualitative, with dramatic shifts in relevant lipid structures. Finally, we compared the results found in Wistar rats with an analysis in a human patient cohort of extreme obesity. For the first time to our knowledge, lipid carbonyl levels and lipoproteins profiles were analyzed in the context of subclinical NAFLD. The association found between lipid carbonyls and alanine aminotransferase (ALT) in a human cohort of extremely obese individuals further supports the potential role of lipid moieties as biomarkers of early NAFLD.

## 1. Introduction

Non-alcoholic fatty liver disease (NAFLD) is a condition in which excess fat builds up in the liver. The overall global prevalence of NAFLD has increased drastically in the last few decades, reaching up to 30.1% [1]. The typical NAFLD profile is a patient who consumes large amounts of fat, sugars, or both, with dramatic consequences in liver biology. These diets are associated with adipose tissue hypertrophy and increased lipolysis, promoting the release of more free fatty acids into the bloodstream and with insulin resistance, upregulating liver de novo lipogenesis and extracellular fatty acids uptake in the liver [2,3].

Fat accumulation in the hepatocytes, called steatosis, may begin as a potentially protective mechanism [4]. Under cellular stress, beta-oxidation of fatty acids is upregulated, and mitochondrial activity may be excessive, which in turn may induce oxidative stress and mitochondrial dysfunction [4]. When mitochondrial beta-oxidation is compromised, omega-oxidation of fatty acids may be triggered, producing more reactive oxygen species (ROS) and dicarboxylic acids toxic to the cell [5,6]. The outcome of this process is thus an increase in lipids in the cell with an imbalance between the different species, and an increase in the oxidized forms such as reactive carbonyls species [7,8,9,10]. These alterations in hepatic fat profiles have a strong impact on blood lipids, causing alterations in the latter that have been linked to NAFLD, although the specific changes in the liver in the early stages of the disease are still unclear [11].

Although epidemiological evidence is still limited and the available information is partially contradictory, sex seems to play a role in NAFLD development [12,13]. Sex-specific fat distribution and differential lipid metabolism could be major players, adding weight to this theory [14]. Additionally, the liver is a highly sexually dimorphic organ in healthy conditions, and gene expression studies show sex differential transcriptomic profiles in both healthy and diseased livers [15]. All of this evidence notwithstanding, sex dimorphism in liver metabolism of NAFLD is a poorly understood area. 

Recent studies revealed dramatic changes in specific fatty acids in liver samples from NAFLD patients, suggesting that dysregulation of lipid metabolism is critical in the pathogenesis and progression of NAFLD [16]. However, strategies focused on changing the composition in these specific fatty acids have not been effective [17]. Methods focused on more general classes of lipids, such as treatment with polyunsaturated fatty acids (PUFAs), have been reported as beneficial for NAFLD patients [18]. Other approaches based on reducing oxidized reactive lipid species in mice have also been suggested [19]. Lipid oxidation produces reactive carbonyls, although the extent depends on composition, level of unsaturation, and oxidative conditions. Reactive carbonyls are toxic mediators of further oxidative damage in the progression of many diseases. 

Metabolomics and lipidomics have boosted the use of spectroscopic techniques in the life sciences. Among these spectroscopic techniques, nuclear magnetic resonance spectroscopy (NMR), in all of its modalities, has some features that make it the best option for many studies. NMR is truly quantitative and can be applied to liquid, semisolid and solid samples without any prior sample treatment or destruction, thus allowing the study of the original physicochemical states of species and moieties. High-resolution magic angle spinning NMR spectroscopy (HR-MAS NMR) is a powerful technique for studying metabolites within different intact tissues [20,21]. Analyzing intact tissues eliminates the need for metabolite extraction, and with it the changes entailed in metabolic composition and properties. HR-MAS used in liver samples reveals accurate information on the composition and properties of lipids and may help us in gaining a greater understanding of lipid metabolism in NAFLD. 

Our study aimed to analyze serum and liver lipid moiety profiles (specifically unsaturations and carbonyls) in a subclinical Wistar rat model of NAFLD to detect relevant alterations and explore potential sex-dependent changes, and thus obtain new insight into lipid metabolism and identify potential sex-specific biomarkers of early fatty liver disease.

## 2. Materials and Methods

### 2.1. Experimental Design and Sampling Method in Wistar Rats

Eighteen-week-old male (499.2 ± 44.5 g) and female (267.2 ± 12.5 g) Wistar rats (Wistar rats, RjHan:WI strain, Janvier Labs, Le Genest-Saint-Isle, France) were stratified by sex and randomly subdivided to receive one of two different diets for 12 weeks: chow diet (2014S, ENVIGO, Indianapolis, IN, USA) or 45% high-fat diet (TD.08811, ENVIGO, Indianapolis, IN, USA). The experimental groups were as follows: *n* = 8 males fed with chow diet (CTL male group), *n* = 8 males fed with high-fat diet (HFD male group), *n* = 8 females fed with chow diet (CTL female group), *n* = 7 females fed with high-fat diet (HFD female group). One of the HFD females presented with altered biochemical parameters and, being considered an outlier, was excluded from the study. The animals had *ad libitum* access to food and drinking water. They were housed in a 12 h light/dark cycle in a room with controlled temperature and humidity. The development of this project was approved by the Ethics Committee for Animal Welfare of the University of Valencia (Spain) (code: A1405675789374). During the procedure, the animals did not undergo any type of intervention or method that required anesthesia. Animal well-being was controlled by the experimenters and supervised by the center’s veterinarian. 

Before the sacrifice, basal glycemia was measured in fasting conditions using a glucometer (Breeze 2, Bayer, Leverkusen, Germany) and glucose test strips (1465A, Bayer, Germany). The drop of blood was obtained by puncture of the saphenous vein of the hind leg. After 12 weeks, the animals were sacrificed by inhalation of 5% isoflurane. All of the blood from the animal was collected, and the serum was obtained by centrifugation at 2000× *g* for 30 min at 4 °C. The liver was weighed, and one small portion was preserved in Tissue–Tek (4583, Sakura, Torrance, CA, USA) and another in formaldehyde 4% (11699455, VWR Q-Path Chemicals, Barcelona, Spain) for future histological analysis. The remaining liver was preserved at −80 °C until metabolomic analysis and MDA determination.

### 2.2. Human Cohort of Extreme Obesity

This was a subset of a case–control study conducted in 1350 obese patients. The body mass index (BMI) was calculated with the weight (kg) and the height (m) of these patients. All of the patients were extremely obese because they presented a BMI ≥ 40 kg/m^2^. The study was carried out at the Division of General Medicine of the San Giuseppe Hospital, Istituto Auxologico Italiano (Oggebbio, Italy). The patients were recruited for diagnostic or therapeutic problems related to obesity or its comorbidities during the period 2009–2010 [22]. The insulin resistance (IR) was checked by the homeostatic model assessment for insulin resistance (HOMA-IR) parameter. It was calculated by applying the following formula: (fasting insulin (mIU/L) × fasting glucose (mg/dL))/22.5. We selected individuals between a borderline IR status (HOMA-IR < 3) and minor grade of IR status (HOMA-IR 3.0–4.99) for the present study, recruiting a total of 264 individuals. All of the voluntary participants were given an informed consent form with all of the information concerning the study prior to taking part. In the session on 10 December 2008, the studies were approved by the Ethics Review Committee of Istituto Auxologico Italiano (Milano, Italy), and all of the participants gave written agreement to participate, including informed consent for their blood samples to be used for research studies. Blood samples were drawn in the early morning following an 8–12 h fasting period. An aliquot of blood was obtained to carry out the routine blood biochemistry analysis. Glucose levels, low-density lipoprotein (LDL), high-density lipoprotein (HDL), triglycerides, alanine aminotransferase (ALT) and total bilirubin (TBIL) were determined in the molecular laboratory of the Division of General Medicine of the San Giuseppe Hospital, Istituto Auxologico Italiano (Oggebbio, Italy). The ALT and TBIL parameters were consulted to create violin plots. A further aliquot of blood was collected in a covered test tube and was left undisturbed at room temperature from 30 min to 1 h to allow the blood to clot. The serum was then separated from the clot by centrifuging at 1000–2000× *g* for 10 min and frozen immediately at −80 °C until the NMR measurements were performed (see Section 2.5). 

### 2.3. Serum Biochemistry Parameters Measured in Wistar Rats

Serum biochemistry parameters in Wistar rats were measured by different commercial kits. Triglycerides analysis was carried out using the Triglyceride Quantification Assay Kit (ab65336, Abcam, Cambridge, UK). HDL, very low-density lipoprotein (VLDL) and total cholesterol analysis was carried out using the HDL and VLDL Quantification kit (MAK045, Sigma-Aldrich, Darmstadt, Germany). An ALT activity assay kit (MAK052, Sigma-Aldrich, Darmstadt, Germany) was used to measure ALT activity in serum. Briefly, all of the kits were based on a color enzymatic reaction measured by a spectrometer (SpectraMax Plus 384 Microplate Reader, Molecular Devices, San Jose, CA, USA) at λ = 570 nm. All kits were used following the manufacturers’ instructions. Additionally, the insulin levels (ng/mL) were measured using the Insulin Enzyme linked immunosorbent assay (ELISA) kit (EZRMI-13K, Millipore, Burlington, MA, USA). Briefly, the assay captured the insulin molecules in the sample by antibodies already bound in the plate. The activity of the enzyme was read spectrophotometrically at λ = 450 nm and λ = 590 nm to calculate the absorbance increase. The insulin levels were measured to determine if the rats could develop IR. 

### 2.4. Liver Histology in Wistar Rats

Liver fragments used for histology were obtained from distal fragments of the larger lobes. Liver samples stored in 4% formaldehyde were embedded in paraffin using a Leica TP1020 Semi-Enclosed Automatic Benchtop Tissue Processor (Leica Microsystems, Wetzlar, Germany). Next, paraffin-embedded tissues were cut with a HistoCore Multicut microtome (Leica Microsystems, Wetzlar, Germany) into 3–5 µm sections and used to perform water-based stains such as hematoxylin and eosin (H&E) (1.09253 and 1.09844, Sigma-Aldrich, Darmstadt, Germany), Masson’s Trichromic (F8129, 861286 and 415049, Sigma-Aldrich, Darmstadt, Germany) and periodic acid–Schiff staining (PAS) (P7875 and 1.09033, Sigma-Aldrich, Darmstadt, Germany). Liver samples embedded in Tissue-Tek were used to carry out a non-water-based stain to specifically dye the fat with Oil Red O (ORO) staining (O0625, Sigma-Aldrich, Darmstadt, Germany). Frozen liver sections were obtained using a Leica CM1900 Cryostat (Leica Microsystems, Wetzlar, Germany). Only the ORO stain was quantified. Five photographs per animal were taken randomly along the liver section using a Leica DMD108 Digital microimaging device (Leica Microsystems, Wetzlar, Germany). These photographs were quantified with the ImageJ2 Software (ImageJ2, software free from: https://imagej.net, accessed on 21 January 2023) by selecting the “Color Threshold” function and applying the following settings for red color detection (210-255 Hue, 130-255 Saturation, 160-255 Brightness). 

### 2.5. Lipid Profiling by ^1^H-NMR and HR-MAS in Wistar Rats and Human Cohort Serum Samples

A metabolomic study of samples was carried out with a Bruker AVANCE III NMR spectrometer (Bruker BioSpin GmbH, Rheinstetten, Germany) operating at a ^1^H frequency of 600.13 MHz. The spectrometer was equipped with different probes to allow the specific measurement of the samples. These probes were the following: (i) a 1 mm ^1^H-^13^C-^15^N triple resonance (TXI) probe, (ii) a 5 mm TXI probe, and (iii) a TXI high-resolution magic angle spinning (HR-MAS) probe. The differences among them were the types of samples that could be measured and the sample volume. Proton nuclear magnetic resonance (^1^H-NMR) was applied only for the detection of protons within the molecules. To the serum samples, we added trimethylsilylpropanoic acid (TSP) (11202, Deutero, Germany), which was used as the NMR standard.

A 1 mm TXI probe was used for the metabolomic measurement of Wistar rat serum. For each sample, a mixture of 20 µL of serum and 2 µL of phosphate buffer with TSP 2.5 mM and deuterium oxide (D_2_O) (1.13366, Sigma Aldrich, Hamburg, Germany) for field locking purposes was transferred into a 1 mm NMR capillary tube (Z107504, Bruker, Rheinstetten, Germany). A single-pulse presaturation experiment was performed in all samples. The number of transients was 256, collected into 65 k data points for all experiments.

A 5 mm TXI probe was used for the metabolomic measurement of human cohort serum samples. A mixture of 470 µL of serum plus 30 µL of TSP 2.5 mM and D_2_O was transferred into a 5 mm NMR capillary tube (Z172600, Bruker, Rheinstetten, Germany). Presaturation spectra were acquired using a 3.95 s acquisition time and 32 transients.

For the metabolomic measurement, Wistar rat livers kept at −80 °C were fractioned by liquid nitrogen. The fragments were weighted (between 40–50 mg) and introduced into a 4 mm zirconia oxide rotor (HZ07213, Bruker, Rheinstetten, Germany) with 40 µL D_2_O. A TXI HR-MAS probe was used for the lipid profile of intact liver tissue. This probe allowed the measurement of intact tissue metabolism. The samples were spun at 5000 Hz to suppress line broadening associated with solid samples.

All of the spectra obtained were processed through the MestReNova software (MestReNova v14.1.1, Mestrelab Research S.L, Santiago de Compostela, Spain). The phase, the baseline, and the reference of the spectra (to the second Alanine peak, 1.478 ppm) were corrected. Spectra were normalized to the aliphatic area (0.5–4.5 ppm). In addition, the liver fragment mass was used to normalize the liver data. Lipoparticle profiles were obtained by in-house scripts and linear regression models calibrated against well validated, previously published methods [23]. Lipid moiety NMR peaks were integrated and quantified using semi-automated in-house MATLAB peak-fitting routines (MATLAB R2014a, MathWorks, Natick, MA, USA). Final lipid structure levels were calculated in arbitrary units as peak area normalized to the total spectral area. Six regions corresponding to different lipid structures were identified and analyzed in the serum (Appendix A) and the liver (Appendix A) spectra as explained elsewhere [21]. The six regions were the following: 1: SFA, saturated fatty acids; 2: lcCO, long chain carbonyl groups; 3: lcUFA, long chain unsaturated fatty acids; 4: tCO, total carbonyl groups; 5: PUFA, polyunsaturated fatty acids; 6: tUFA, total unsaturated fatty acids. Chemical structure assignments were confirmed using a total correlation spectroscopy (TOCSY) experiment in both serum and liver. Relative levels of lipid regions were expressed as the ratio of the region spectral area to total lipid spectral area for each individual spectrum. 

### 2.6. MDA Determination in Wistar Rats 

Malondialdehyde (MDA) was analyzed in serum and liver extract samples. For serum, an aliquot of 100 µL was needed. For liver extract, frozen livers were fractionated by liquid nitrogen. A fragment of 90–100 mg was homogenized in potassium phosphate buffer (Kpi) 50 mM (100 µL of buffer per 10 mg of tissue), ethylenediaminetetraacetic acid (EDTA) 1 mM, pH 7.4 using an Ultra-Turrax (IKA, Königswinter, Germany) and an Ultrasonic processor 500 Watt (Sonics & Materials INC, Newtown, CT, USA). Then, the liver homogenate was centrifuged at 500× *g* for 5 min at 4 °C. The supernatant was collected. An aliquot of at least 100 µL of supernatant was needed for the following steps. The protein quantity was quantified in the supernatant using the Pierce BCA protein assay kit (23225, ThermoFisher, Waltham, MA, USA). We used the mg of protein in the supernatant to further normalize the data. Serum and liver extract samples were derivatized by mixing the samples in 500 µL of sodium acetate (NaOAc) 2M pH 3.5 and 0.2% of thiobarbituric acid (TBA) before measurement by ultra performance liquid chromatography (UPLC). The results for MDA in serum were expressed in µM, and the results for MDA in liver extract were expressed in nmol of MDA/mg of liver protein.

### 2.7. Statistical Analysis

Statistical analysis was carried out using SPSS Statistics (IBM SPSS Statistics 26, New York, NY, USA). Between-group differences were evaluated by different tests. First, the normality of the variables was checked with the Shapiro–Wilk test. Next, homogeneity of the variances was checked with the Levene test, and a one-way analysis of variance (ANOVA) test was carried out. Post hoc Bonferroni or Games–Howell tests were applied depending on whether the variables had homogeneity of variances or not, respectively. Only one variable (ORO staining) did not have a normal distribution. For these, the non-parametric Kruskal–Wallis test was applied. The data were reported as the mean ± standard deviation for all parameters. Statistically significant influences of diet were represented by (*), while statistically significant influences of sex were represented by (^†^). Statistical significance was set at *p* < 0.05. The cluster-arranged heatmap, the Circos plots, the boxplots, the z-score plots, the correlation matrices, and the violin plots were drawn in the R software environment (version 4.1.3). Statistical analysis of these plots was conducted also using R software. In addition, for the Circos plots, the plugins Circlize and EpiViz were used. Again, statistical significance was set at *p* < 0.05.

## 3. Results

### 3.1. Twelve Weeks of HFD Induced Subclinical Fat Accumulation in Wistar Rat Liver, with Minor Metabolic Alterations Observed Only in Male Wistar Rats

Twelve weeks of an HFD had a greater effect on male than on female Wistar rats (Table 1), resulting in a statistically significant increase in body mass in male Wistar rats only. This increase was paralleled by a proportional rise in liver mass. However, the liver mass/body mass ratio showed no statistically significant differences between CTL and HFD animals regardless of sex, suggesting that liver mass gains were due only to the proportional increase in the size of the animal. We observed significant differences in body mass, liver mass, and the liver mass/body mass ratio between male and female Wistar rats in both CTL and HFD animals. This was expected, since female Wistar rats are much smaller than their male counterparts. 

Masson’s Trichrome and PAS stains showed no differences between experimental groups (Figure 1a). There were no signs of fibrosis or changes in glycogen storage pattern in the tissue between CTL and HFD groups, or male and female samples. However, H&E staining revealed incipient steatosis in HFD male and female Wistar rats that was confirmed by specific fat staining with ORO stain (Figure 1a). Accumulation of fat and liver steatosis was further validated by quantification of the ORO stain (Figure 1b). A similar, significant increase in fat was observed in both male and female Wistar rats in HFD groups compared to CTL groups. There were no statistically significant differences between male and female Wistar rats for any of the histological parameters examined. 

Statistically significant differences in serum biochemical parameters were found only between CTL and HFD male groups (Table 1). HDL levels were lower in HFD males compared to CTL males. Triglyceride levels and basal glycemia seemed to be higher in HFD males than in CTL males, but no statistically significant differences were observed. Female Wistar rats did not show statistically significant differences between CTL and HFD animals in any serum biochemical parameter. Several serum biochemistry parameters showed significant differences between sexes, including triglycerides, VLDL levels, and insulin levels. Focusing on insulin values and basal glycemia, neither males nor females developed insulin resistance (IR).

### 3.2. Twelve Weeks of HFD Induces Changes in Blood Lipoparticles and Lipid Unsaturation in Wistar Rats

A change in the relative levels of the different lipid structures detected by NMR was observed after 12 weeks of HFD. In general, changes were greater in male than in female Wistar rats (Figure 2a). Lipoparticle profiles in serum were affected by 12 weeks of HFD, exhibiting global increases in particle number except for HDL particles in male and female Wistar rats. The main increases were reflected in triglycerides content, in VLDL particles and LDL particles, with some decreases in esterified cholesterol (CE) in HDL particles (significant differences marked with an empty circle) (Figure 2a). The cluster-arranged heatmap of the serum lipid spectral region levels for each animal (Figure 2b) showed decreases in unsaturations and increases in carbonyl content in lipid after 12 weeks of HFD, with strong effects in males and moderate to negligible effects in females. After 12 weeks of HFD, female Wistar rats showed higher levels of carbonyls in long-chain fatty acids (lcCO), long-chain unsaturated fatty acids (lcUFA) and polyunsaturated fatty acids (PUFAs), whereas male Wistar rats had higher levels of saturated fatty acids (SFAs), total carbonyls in lipids (tCO), and total UFAs (tUFA). CTL and HFD male Wistar rats showed evident differences. HFD male rats presented lower levels of PUFAs and tUFA and slightly higher levels of SFA and tCO compared to CTL males. Focusing on CTL and HFD females Wistar rats, the HFD females presented lower levels of lcUFAs and slightly higher levels of tCO and lcCO compared to CTL females. Lipid peroxidation measured by MDA did not show statistically significant differences associated with HFD in any group (Figure 2b). The 90% confidence intervals (CI) of the z-scores of the different moieties for each group (Figure 2c) further supported the changes observed in the heatmap and demonstrated that although following the same direction as male Wistar rats, most changes observed in female Wistar rats were of lesser intensity. The exception to this trend was carbonyl groups in lipids (tCO and lcCO), in which the change was more pronounced in females than in male Wistar rats. Boxplots of tCO (Figure 2d) showed statistically significant differences only between CTL and HFD male Wistar rats (*p* = 0.007 < 0.05), although the boxplots displayed high overlapping. However, lcCO (Figure 2e) showed significant differences between HFD male and HFD female Wistar rats (*p* = 0.0093 < 0.05), with lcCO levels being higher in HFD females. Interestingly, in both spectral regions, the dispersion of values was higher in female rats than in male rats, with wider boxes and larger standard deviations.

### 3.3. Total Lipid Unsaturations Are Altered in a Sex-Dependent Manner, but Lipid Carbonyl Changes Are Sex-Independent in Wistar Rat Liver after HFD

The relative contents of lipid unsaturations and lipid carbonyl species in liver were measured using HR-MAS ^1^H-NMR spectroscopy. Lipid structure levels were quantified as the ratio between the spectral area of the region containing the lipid structure resonance and the total spectral area for lipid in the spectrum. The changes in these lipid structures were of larger intensity than those detected in the serum, as expected in a subclinical fatty liver context (Figure 3). Overall, the stronger effect was observed in lipid carbonyls, both tCO and lcCO, in both sexes. There was also a general decrease in liver lipid unsaturations (in both PUFAs and lcUFAs) after 12 weeks of HFD. The clustered heatmap (Figure 3a) revealed higher relative levels of lipid structures in SFAs, lcCO, tUFAs, and tCO after 12 weeks of HFD in HFD groups. However, the relative levels for these lipid structures were higher in HFD male than in HFD female Wistar rats. MDA seemed to experience a moderate decrease after 12 weeks of HFD, but this could be related to the decrease in unsaturations. In addition, the heatmap showed slightly different effect patterns between female and male Wistar rats. CTL animals showed higher levels of PUFAs and lcUFAs. These CTL groups showed great similarities between male and female Wistar rats. However, 12 weeks of HFD produced stronger effects and diverse patterns in males, whereas female Wistar rats exhibited a less robust and defined pattern. Again, the 90% CI of the z-scores of the different moieties for each group (Figure 3b) further supported the changes observed in the heatmap and the lower variation in effect intensity between male and female Wistar rats than in serum. The boxplots showing the distribution of lipid carbonyls levels (tCO and lcCO) in the different groups (Figure 3c and 3d, respectively) presented significant differences between CTL and HFD in both male (*p* = 0.0011 < 0.05, for tCO and lcCO) and female (*p* = 0.0059 < 0.05 for tCO; *p* = 0.0093 < 0.05 for lcCO) Wistar rats with no between-groups overlapping. Moreover, there were statistical differences between HFD males and HFD females in tCO (*p* = 0.021 < 0.05) and lcCO (*p* = 0.029 < 0.05) levels, suggesting a key role for lipid carbonyl metabolism in liver disease development.

### 3.4. HFD Dysregulates the Interaction between Lipid Unsaturations, Fatty Acid Carbonyl Content and Lipid Peroxidation by MDA Quantification

MDA is a reactive molecule formed as a byproduct of lipid peroxidation, a process in which UFAs in cell membranes are attacked by ROS, leading to lipid degradation. To further explore the interactions between oxidative stress and lipid moieties, MDA was quantified in serum and liver extract (Table 2). There was no statistically significant increase in serum MDA values among the experimental groups. Nevertheless, there was a statistically significant decrease in liver MDA values of HFD males. Moreover, the difference between HFD males and HFD females was also statistically significant. As the only statistically significant differences were seen in liver MDA values, these liver MDA values were represented in a boxplot (Figure 4a). To explore the potential effect of unsaturation amounts on measurements, we normalized the liver MDA value to total liver unsaturations (MDAnorm) (Figure 4b), observing a statistically significant (*p* = 0.021 < 0.05) increase in liver MDAnorm of HFD males compared to CTL males. Next, serum MDAnorm was calculated, but there were still no differences after 12 weeks of HFD, suggesting a potential increase in lipid peroxidation associated with subclinical fatty liver. The correlation matrices between liver lipid moieties and liver MDA values for each group showed a reduction in the amount and number of significant correlations after HFD, except for MDA correlations in female rats after 12 weeks, which became stronger (Figure 4c). Although the interpretation of these results is complex, HFD appears to induce dysregulation of lipid metabolism in male rats, whereas females have some kind of protection mechanism that helps maintain these correlations.

### 3.5. Carbonyls Analyzed in Blood of an Obese Human Patient Cohort Are Associated with Liver Biochemical Parameters

The characterization of the Piancavallo cohort composition and metabolic characteristics appears in Table 3. The obese patient population (BMI ≥ 40 kg/m^2^) was around 54.5 ± 14.4 years old, and it was composed of a higher proportion of females (*n* = 192) than males (*n* = 72). There were not significant differences in biochemical parameters (glucose, LDL, HDL, triglycerides, ALT and TBIL). The selected population was in a borderline or minor grade of IR status. Given that these patients came from a cohort of extremely obese patients, we considered the selected patients for this study as not insulin resistant. Carbonyls showed statistically significant differences in both the serum and liver samples from Wistar rats measured by NMR. Due to their importance and their potential to be used as biomarkers of changes in lipid moieties and lipid structures, carbonyls were analyzed in the human cohort of obese patients to determine the association of their tertiles with liver function biochemical parameters (ALT and TBIL) (Figure 5). We found that obese individuals in the highest tertile of lcCO also exhibited moderately high levels of ALT (Figure 5a) and total bilirubin (TBIL) (Figure 5b). 

## 4. Discussion

Dietary and/or genetic methods can be used to induce metabolic disease in animal models. An HFD can produce NAFLD, although it often also induces insulin resistance and other metabolic syndrome symptoms, making it difficult to isolate this condition. In this study, we fed 18-week-old male and female Wistar rats an HFD for 12 consecutive weeks. After this dietary intervention, the rats in the HFD group showed fat accumulation in the liver and exhibited increased body mass, but no increase in the liver mass/body mass ratio compared with levels in the age-matched control rats. No liver dysfunction was observed, confirming the subclinical condition of fatty liver. In addition, there were no statistically significant differences in basal glycemia and insulin values; therefore, no insulin resistance was present. Twelve weeks of HFD also caused changes in the blood triglycerides and lipoproteins and in liver fat accumulation, suggesting that subclinical NAFLD was successfully induced in Wistar rats. Our Wistar rat model seems suitable for the study of subclinical and early NAFLD, successfully isolating it from other concomitant conditions. This is the first time to our knowledge that lipid moieties, lipoprotein profiles, and lipid peroxidation have been analyzed together and in relation to sex in subclinical NAFLD. Lipid moieties are the building blocks of lipids, and their analysis can provide information on the biosynthesis, degradation, and remodeling of lipids. Additionally, studying lipid moieties can help identify new biomarkers for various diseases.

Studies of lipid profiles for identifying the differences between males and females are still scarce. Herein, we focused on determining sex-specific lipid moiety profiles in a model of subclinical NAFLD in Wistar rats to identify potential new biomarkers and explore the underlying mechanisms. Although our rats developed liver steatosis regardless of sex, dramatic changes in relevant lipid structures were produced in a sex-dependent manner. Although recent studies have reported sex-based effects in the production of fatty acid derivatives [24], ours is the first to analyze key lipid structures as related to sex. We found that 12 weeks of HFD induces fat accumulation in the liver to a similar extent in male and female Wistar rats. Similarly, this fat accumulation paralleled an increase in lipid carbonyl species and a decrease in PUFAs, in both circulating blood and in liver. However, the intensity of changes in composition (i.e., carbonyls and PUFAs) was much lower in female than in male Wistar rats. Additionally, an increase in circulating triglycerides and a decrease in VLDL levels suggested liver metabolism alterations in male but not in female Wistar rats after 12 weeks of HFD. Although our findings are in line with previously reported female protection against NAFLD, we have pinpointed a novel association with specific lipid moieties related to lipid oxidation.

Fatty acids can reach the liver in three different ways: (1) from adipose tissue (AT) reserves, (2) via blood after ingestion or (3) from remnants of lipoproteins. Hepatocyte fatty acids not immediately consumed are esterified to be stored as triglycerides, in the form of lipid droplets that are used as cellular reserves or can be mobilized through VLDL lipoproteins [25]. Our study revealed major HFD-induced changes in the lipoparticle profile, including triglyceride content and VLDL particle composition and metabolism. Although we observed subclinical fat deposits in both female and male Wistar rat livers, male rats exhibited much stronger alterations in blood lipoproteins and triglycerides after HFD, suggesting a deeper involvement of NAFLD beyond excess of lipids in the hepatocytes. The biochemistry of blood lipids is closely related to liver function and is highly complex. VLDL synthesis is a complex process mostly regulated by ApoB100 protein synthesis and accumulation of triglycerides in the hepatocytes. Once synthesized, VLDLs leave the hepatocytes and enter general circulation, where they export fat to other organs. However, impaired VLDL formation can trigger steatosis development [26,27]. Our observation of high blood triglycerides and low HDL in HFD male Wistar rats alone suggests that fat accumulation-related changes in liver lipid metabolism could be sex-dependent. 

Circulating free fatty acids are normally converted to triglycerides in the liver and stored in adipose tissue, the liver, and other tissues. An excess of circulating fatty acids, therefore, results in increased body and fat mass. Conversion of free fatty acids to triglycerides depends on the properties of hydrocarbon chains, including unsaturations and carbonyl groups. Our NMR analysis of the serum lipid profile revealed a relative decrease in the amounts of PUFAs and carbonyls in lipids in both female and male Wistar rats. Although lipoprotein levels were not affected by HFD in females, the composition of circulating fatty acids shifted to partially resemble the profile observed in males. This suggests that the alterations in liver lipid metabolism induced by an HFD are common to both sexes but compensated by sex-specific mechanisms in females. 

Carbonyl groups are functional groups consisting of a carbon atom double-bonded to an oxygen atom (C=O), and neighboring protons can be specifically detected and quantified by NMR. In our study, we used this approach to detect a robust increase in carbonyl content in both total (tCO) and in long chain lipids (lcCO) after 12 weeks of HFD in Wistar rats. Carbonyl groups in long-chain fatty acids (lcCO) are essential for energy metabolism, particularly in the context of beta-oxidation and ATP production. They allow controlled breakdown of fatty acids into usable energy units. Furthermore, carbonyl groups present in the total fatty acid pool (tCO) contribute to a wide range of physiological processes, including cell membrane formation, lipid signaling, and the synthesis of biologically active molecules. The changes observed in carbonyl content in serum lipids in Wistar rats suggest a potential role of this moiety as an early disease biomarker. To further support this hypothesis, we measured tCO and lcCO in an extremely obese human cohort. From the 1350 obese patients in the obese human cohort, only 264 individuals were recruited for this study. As the Wistar rats did not present insulin resistance, we chose the individuals with a lower HOMA-IR ratio. The selected individuals were considered as having no insulin resistance, so the comparison with the rats was equivalent. We analyzed the association of their tertiles with biochemical parameters of liver function in this obese human cohort. We confirmed that there was an association between lcCO and high levels of ALT and TBIL in the blood of obese patients, indicating the potential value of carbonyl content in fatty acids for improved characterization of NAFLD.

Interestingly, despite changes in PUFAs and carbonyls, we did not observe clear differences in lipid peroxidation as measured by MDA determination. The link between unsaturations in fatty acids and MDA quantification lies in the process of lipid peroxidation. Unsaturated fatty acids are vulnerable to oxidation, leading to the formation of reactive MDA molecules. MDA originates from PUFAs, when a carbon–carbon double bond is attacked by a free radical, resulting in the formation of an unsaturated lipid radical with H_2_O release. Consequently, MDA levels are difficult to interpret in the context of unstable PUFA content. 

Many studies have demonstrated altered circulating fatty acid profiles in patients with NAFLD. Most studies focus on measuring specific lipid species (for example, palmitates or linoleates) that are closely linked with specific diet patterns. However, biochemical pathways in lipid metabolism work over lipid structures (double bonds in cis or trans configurations, polyunsaturated regions, carbonyls), which may be more useful for understanding the overall mechanisms of disease. Total saturated fatty acids were found to be increased in liver biopsies of patients with NAFLD [28]. Total monounsaturated fatty acids were also elevated in the liver and plasma of NAFLD patients [29]. Our study shows similar trends, although the changes are sex dimorphic, especially in terms of the extent of PUFA decreases after HFD. Carbonyl groups in lipids were also altered in Wistar rats after 12 weeks of HFD. In our study, changes in liver lipid profiles after an HFD were very similar between males and females, except for PUFAs. Liver MDA revealed differences in lipid peroxidation after HFD only for male Wistar rats. As previously mentioned, interpreting lipid peroxidation levels in the context of a lipid-changing environment is highly complex. However, the differences in the results between sexes, together with the findings from blood lipid analysis, could further support a potential compensatory sex-dependent mechanism. It is well-known that estrogens play a protective role in females through antioxidant mechanisms [30,31]. Nonetheless, sex differences in adipocyte morphological and metabolic properties may also play a role in cardioprotection, and in the effects observed in our study. Interestingly, the cluster analysis of the most significant variables of this study shows that males undergo a robust pattern of changes after an HFD, whereas females exhibit a moderate, intermixed and more diverse collection of changes.

There are several limitations to the current study. Models of subclinical NAFLD are very scarce because they constitute a challenge. The histopathological and biochemical changes that occurred in animals under high fat diet are in many cases different from those that happen in humans. On the other hand, although NMR provides very important structural information that allows identifying chemical moieties, it is inherently less sensitive than other techniques such as mass spectrometry, and some species could be missing in our study. The human data were obtained in an observational study, and our data do not provide information about causality mechanisms. In addition, all individuals were of Western European descent, and it is difficult to extrapolate these data to other populations. Finally, we used transaminases and bilirubin levels as indicators of liver function. It is important to realize that these markers are affected by some other conditions. 

## 5. Conclusions

This study showed the sex dimorphic profile of fatty acids in the serum and liver in a subclinical NAFLD Wistar rat model. We observed that the bulk accumulation of fat in hepatocytes after 12 weeks of HFD, as detected by ORO histological stain analysis, was higher in HFD animals. Our lipid profile analysis by ^1^H-NMR revealed that the change in fat is not only quantitative but also qualitative, with shifts in overall lipid composition. Furthermore, we detected underlying differences between male and female Wistar rats in the unsaturations and carbonyl content of these lipids. Overall, our results not only demonstrate that HFD induces subclinical NAFLD in Wistar rats in a sex dimorphic manner, but may also help in identifying sex-specific biomarkers of disease. Analysis of the obese human patient cohort revealed potentially relevant associations that further support the findings in animals. Lipid species analysis could be used as a predictive biomarker; thus, further studies are warranted to reach a more comprehensive understanding of the underlying mechanisms. 

## Figures and Tables

**Figure 1 antioxidants-12-01808-f001:**
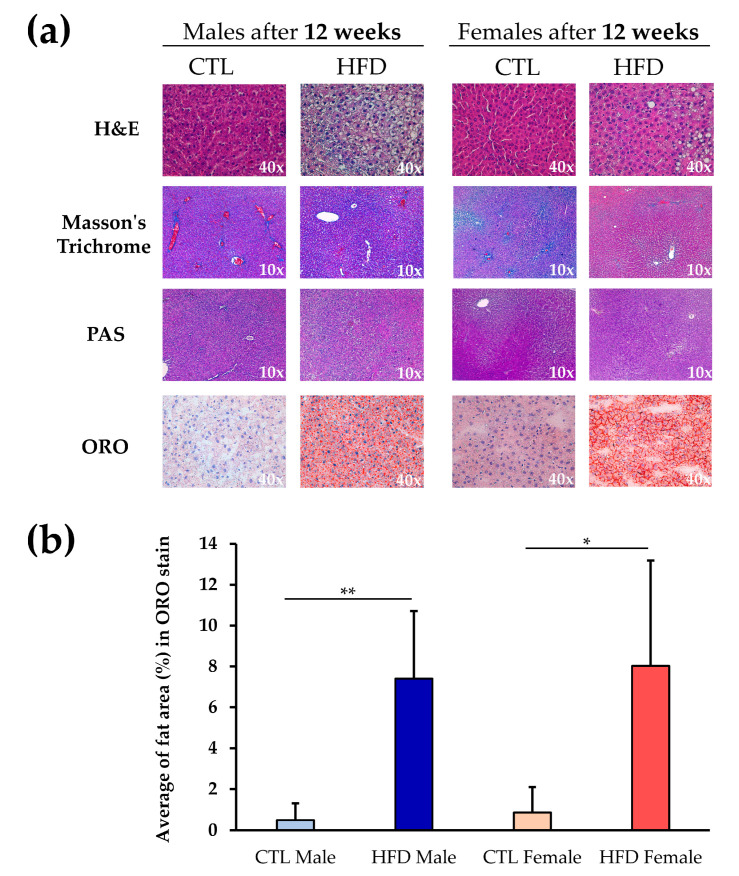
Liver histology in Wistar rats. (**a**) Liver sections stained with H&E, Masson’s Trichrome, PAS, and ORO stains at 10× or 40× magnification in male and female Wistar rats after 12 weeks of CTL or HFD diets. (**b**) ORO stain quantification with ImageJ2 software. Each bar represents the average fat area (%) in each experimental group. Significant differences in ORO stain were calculated by non-parametric Kruskal–Wallis test, * *p* < 0.05; ** *p* < 0.01 between CTL and HFD groups. Abbreviations: CTL, chow diet control group; H&E, hematoxylin and eosin; HFD, high-fat diet group; PAS, periodic acid–Schiff; ORO, oil red O.

**Figure 2 antioxidants-12-01808-f002:**
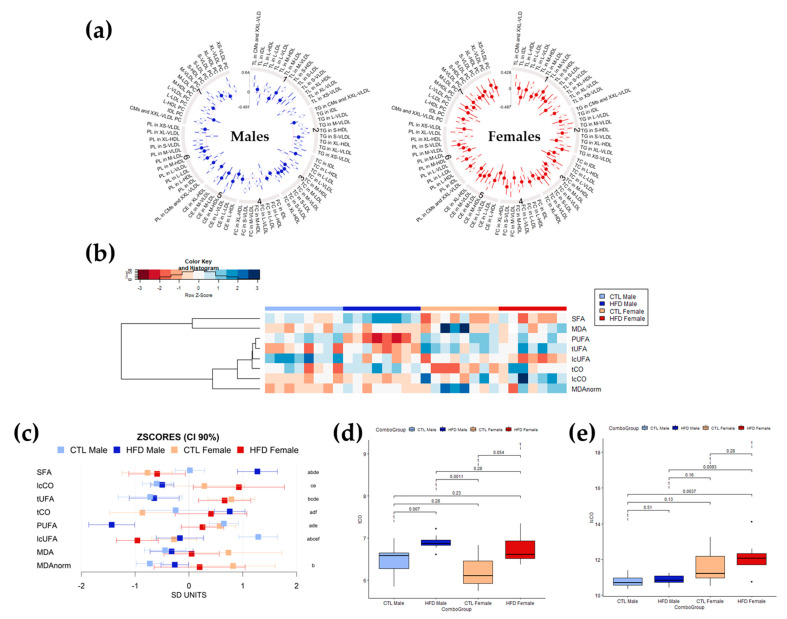
Profiling of lipoparticles and lipid structures in serum determined by NMR spectroscopy after 12 weeks of HFD. (**a**) Circos plot representing the mean differences for lipoparticle parameters in standard deviation units between CTL and HFD male (left, in blue) and female (right, in red) Wistar rats (filled and empty circles) and 95% confidence intervals (radial lines around the circle). Values closer to the center of the plot than the red line represent values higher for CTRL diet than HFD diet, whereas values farther from the center of the plot than the red line represent the opposite trend. Mean significant differences (*p*-value < 0.05) are represented with an empty circle, whereas non-significant mean differences are represented with a filled circle. (**b**) Heatmap and hierarchical cluster analysis of serum lipid moieties and MDA values (rows) for each sample (columns) showing which parameters have a similar trend for all samples in a group. Data have been mean-centered and normalized to the standard deviation for representation. Values higher than the mean appear in red, and those lower than the mean in blue. (**c**) Mean values and 90% CI expressed in SD units for lipid moieties and MDA in the different experimental groups. Multiple pairwise tests with statistical significance at *p* < 0.05 corrected for between-groups comparisons were labelled according to the following code: a, CTL male vs. HFD male; b, CTL male vs. CTL female; c, CTL male vs. HFD female; d, HFD male vs. CTL female; e, HFD male vs. HFD female; and f, CTL female vs. HFD female. (**d**,**e**) Box plots of circulating serum levels for carbonyls in (**d**) tCO and (**e**) lcCO showing specific differences and sample distribution between groups. Boxes denote interquartile range, lines denote median, and whiskers denote tenth and ninetieth percentiles. Statistically significant differences were established at *p* < 0.05. Abbreviations: CE, cholesteryl ester or esterified cholesterol; CI, confidence interval; CMs, chylomicrons; CTL, chow diet control group; HFD, high-fat diet group; lcCO, long chain carbonyl groups; lcUFA, long chain unsaturated fatty acids; FC, free cholesterol; HDL, high-density lipoproteins; IDL, intermediate-density lipoproteins; L, large size; LDL, low-density lipoproteins; M, medium size; MDA, malondialdehyde; MDAnorm, malondialdehyde data normalized; PC, phosphatidylcholine; PL, phospholipid; PUFA, polyunsaturated fatty acids; S, small size; SFA, saturated fatty acids; TC, total cholesterol; tCO, total carbonyl groups; TG, triglyceride; TL, total lipids; tUFA, total unsaturated fatty acids; VLDL, very-low density lipoproteins; XL, extra-large size; XS, extra-small size; XXL, extra-extra-large size.

**Figure 3 antioxidants-12-01808-f003:**
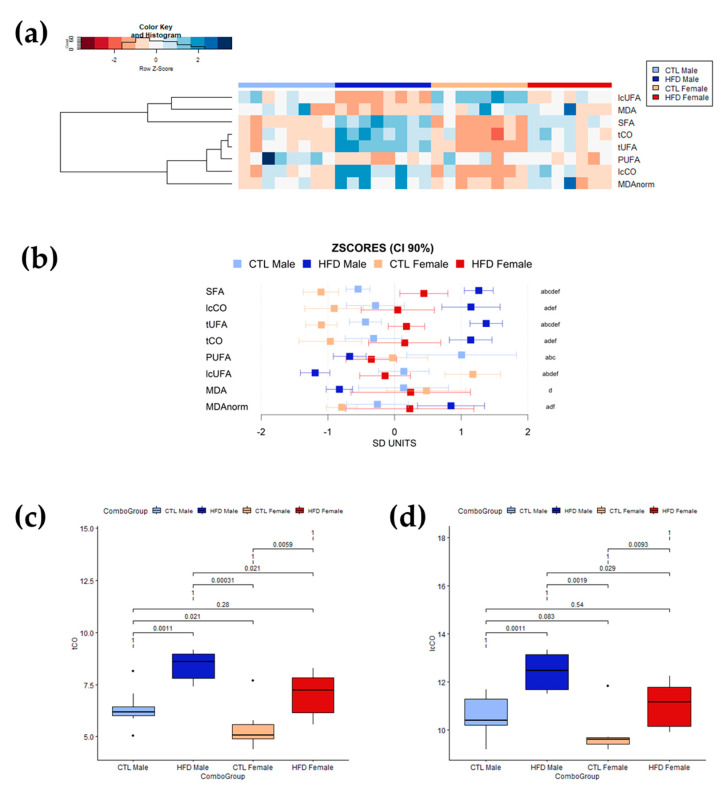
Profiling of lipid structures in intact liver tissue determined by HR-MAS ^1^H-NMR. (**a**) Heatmap and hierarchical cluster analysis of liver lipid moieties and MDA values (rows) for each sample (columns) showing which parameters have similar trends for all samples in a group. Data has been mean-centered and normalized to the standard deviation for representation. Values higher than the mean appear in red, and those lower than the mean in blue. (**b**) Mean values and 90% CI expressed in SD units for lipid moieties and MDA in the different experimental groups. Multiple pairwise tests with statistical significance at *p* < 0.05 corrected for between-group comparisons have been labelled according to the following code: a, CTL male vs. HFD male; b, CTL male vs. CTL female; c, CTL male vs. HFD female; d, HFD male vs. CTL female; e, HFD male vs. HFD female; and f, CTL female vs. HFD female. (**c**,**d**) Box plots of circulating liver levels for carbonyls in (**c**) tCO and (**d**) lcCO showing specific differences and sample distribution between groups. Boxes denote interquartile range, lines denote median, and whiskers denote tenth and ninetieth percentiles. Statistically significant differences were set at *p* < 0.05. Abbreviations: CI, confidence intervals; CTL, chow diet control group; HFD, high-fat diet group; lcCO, long chain carbonyl; lcUFA, long chain unsaturated fatty acids; MDA, malondialdehyde; MDAnorm, malondialdehyde data normalized; PUFA, polyunsaturated fatty acids; SFA, saturated fatty acids; tCO, total carbonyl; tUFA, total unsaturated fatty acids.

**Figure 4 antioxidants-12-01808-f004:**
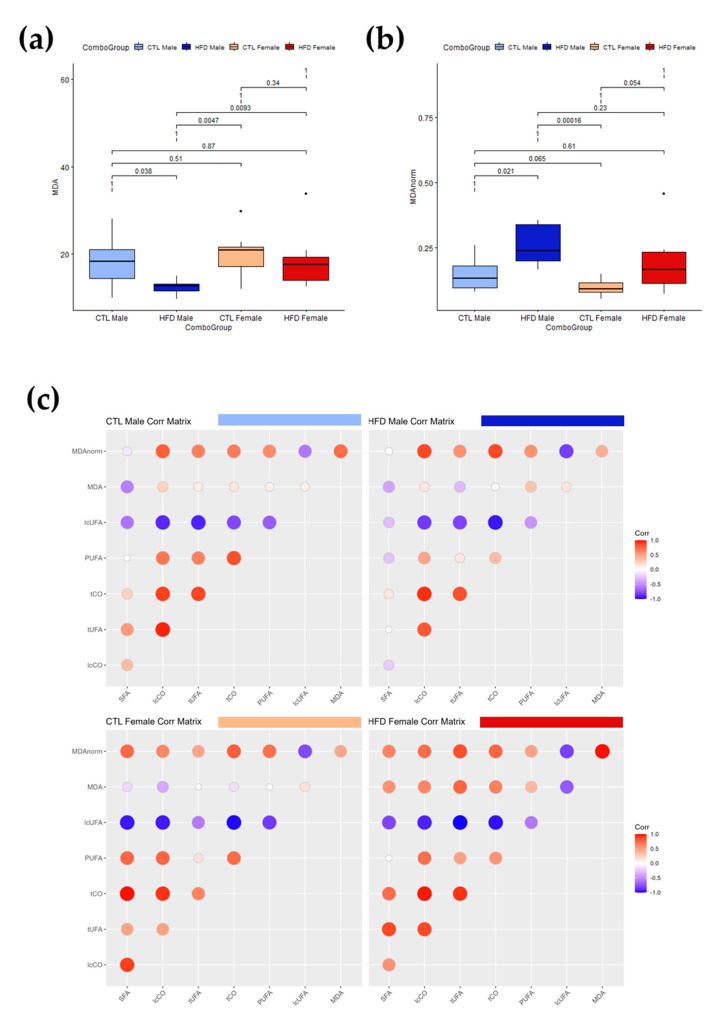
Interactions between MDA, lipid carbonyls and PUFAs in liver samples after 12 weeks of HFD. (**a**,**b**) Box plots of liver levels for MDA without (**a**) and with (**b**) normalization (MDA norm) with respect to total lipid unsaturations. Boxes denote interquartile range, lines denote median, and whiskers denote tenth and ninetieth percentiles. Statistically significant differences were set at *p* < 0.05. (**c**) Correlation matrices between MDA and lipid structures for the different experimental groups showing the loss of correlations induced by HFD after 12 weeks. Stronger correlations are represented by larger and darker color. Negative correlations are represented in blue whereas positive correlations are represented in red. The change in intensity and size or color of the circle represents a change in the correlation which may be related to metabolic dysregulation. Abbreviations: CTL, chow diet control group; HFD, high-fat diet group; lcCO, long chain carbonyl; lcUFA, long chain unsaturated fatty acids; MDA, malondialdehyde; MDAnorm, malondialdehyde data normalized; PUFA, polyunsaturated fatty acids; SFA, saturated fatty acids; tCO, total carbonyl; tUFA, total unsaturated fatty acids.

**Figure 5 antioxidants-12-01808-f005:**
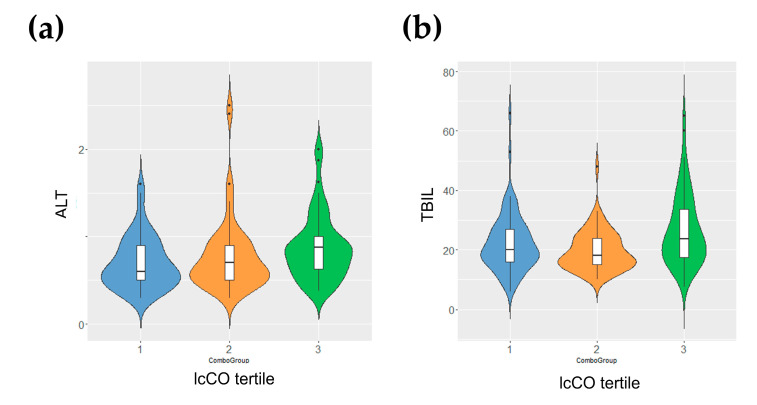
Carbonyl levels in blood from an obese human patient cohort. Levels of (**a**) ALT and (**b**) TBIL for individuals arranged by tertiles (1, left and blue; 2, middle and orange; 3, right and green) of carbonyl content in long chain blood lipids for the subset of extreme obesity human cohort. Tertiles in ALT (1, 22.3 ± 9.9; 2, 19.7 ± 6.7; 3, 26.6 ± 11.9) were statistically significant (*p* = 0.049 < 0.05). Tertiles in TBIL (1, 0.70 ± 0.29; 2, 0.70 ± 0.30; 3, 0.77 ± 0.42) were statistically significant (*p* = 0.047 < 0.05). Abbreviations: ALT, alanine aminotransferase; lcCO, long chain carbonyl; TBIL, total bilirubin.

**Table 1 antioxidants-12-01808-t001:** Metabolic parameters after 12 weeks of HFD in male and female Wistar rats.

Parameter	Males		Females	
	CTL (*n* = 8)	HFD (*n* = 8)	CTL (*n* = 8)	HFD (*n* = 7)
Body mass (g)	633.1 ± 56.6	736.25 ± 93.4 **	297.0 ± 18.8 ^†††^	322.3 ± 28.3 ^†††^
Liver mass (g)	18.5 ± 3.3	21.5 ± 4.1	7.4 ± 0.7 ^†††^	7.7 ± 0.8 ^†††^
Liver mass/Body mass ratio	2.9 ± 0.3	2.9 ± 0.3	2.5 ± 0.2	2.4 ± 0.4 ^†^
Basal glycemia (mg/dL)	86.1 ± 6.0	96.0 ± 8.0	87.5 ± 6.4	88.7 ± 12.6
HDL (mg/dL)	151.4 ± 42.2	98.8 ± 18.4 *	148.3 ± 30.2	132.1 ± 33.1
VLDL (mg/dL)	25.5 ± 6.2	18.00 ± 6.2	17.3 ± 5.1 ^†^	12.7 ± 5.6
Triglycerides (mg/dL)	116.7 ± 29.8	154.2 ± 63.5	58.3 ± 19.3 ^††^	52.2 ± 25.0 ^†^
ALT (mU/mL)	11.2 ± 7.7	4.9 ± 2.2	11.7 ± 4.6	9.3 ± 3.3
Insulin (ng/mL)	3.4 ± 1.3	4.7 ± 1.7	1.1 ± 0.9 ^†††^	2.1 ± 1.0 ^†††^

Data are expressed as the mean values ± standard deviation of the different experimental groups (CTL males *n* = 8; HFD males *n* = 8; CTL females *n* = 8, HFD females *n* = 7). Significant differences were calculated by ANOVA and post hoc test, and set at * *p* < 0.05; ** *p* < 0.01 between CTL and HFD groups; ^†^
*p* < 0.05; ^††^
*p* < 0,01; ^†††^
*p* < 0.001 between CTL males and CTL females or HFD males and HFD females. Abbreviations: ALT, Alanine Aminotransferase; CTL, chow diet control group; HDL, high-density lipoprotein; HFD, high-fat diet group; VLDL, very low-density lipoprotein.

**Table 2 antioxidants-12-01808-t002:** MDA quantification in serum and liver extract after 12 weeks of HFD.

Parameter	Males		Females	
	CTL (*n* = 8)	HFD (*n* = 8)	CTL (*n* = 8)	HFD (*n* = 7)
MDA in serum (µM)	15.99 ± 5.15	17.09 ± 5.35	26.60 ± 13.40	20.44 ± 5.61
MDA in liver extract (nM MDA/mg protein)	0.20 ± 0.06	0.12 ± 0.02 *	0.21 ± 0.06	0.20 ± 0.06 ^†^

Data are expressed as the mean values ± standard deviation of the different experimental groups (CTL males *n* = 8; HFD males *n* = 8; CTL females *n* = 8, HFD females *n* = 7). Significant differences were calculated by ANOVA and post hoc test, and set at * *p* < 0.05 between CTL and HFD groups. ^†^
*p* < 0.05 between CTL males and CTL females or HFD males and HFD females. Abbreviations: CTL, chow diet control group; HFD, high-fat diet group; MDA, malondialdehyde.

**Table 3 antioxidants-12-01808-t003:** Human cohort characteristics and comparison of metabolic characteristics (profile and components) in the entire cohort (total), and in male and female patient subsets.

Variables	Total (*n* = 264)	Male (*n* = 72)	Female (*n* = 192)
Age (years)	54.5 ± 14.4	52.4 ± 13.4	56.4 ± 15.6
BMI (kg/m^2^)	48.7 ± 6.5	48.3 ± 7.6	47.6 ± 6.8
HOMA-IR	3.3 ± 2.7	3.6 ± 2.8	3.0 ± 2.3
Glucose (mg/dL)	102 ± 35	102 ± 34	100 ± 37
LDL Cholesterol (mg/dL)	120 ± 35	121 ± 32	125 ± 39
HDL Cholesterol (mg/dL)	44 ± 14	36 ± 10	42 ± 15
Triglycerides (mg/dL)	147 ± 64	153 ± 62	139 ± 60
ALT (U/L)	23 ± 10	27 ± 10	19 ± 7
TBIL (mg/dL)	0.72 ± 0.34	0.78 ± 0.34	0.67 ± 0.34

Abbreviations: ALT, alanine aminotransferase; BMI, body mass index; HDL, high-density lipoprotein; HOMA-IR, homeostatic model assessment for insulin resistance; LDL, low-density lipoprotein; TBIL, total bilirubin.

## Data Availability

The data presented in this study are available on request from the corresponding author. The data from the human cohort are not publicly available for ethical reasons.

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
