# Peer review of "Altered Lipid Moieties and Carbonyls in a Wistar Rat Dietary Model of Subclinical Fatty Liver: Potential Sex-Specific Biomarkers of Early Fatty Liver Disease?"

_antioxidants, 2023, doi:10.3390/antiox12101808_

Round 1
Reviewer 1 Report
The authors investigated lipid moieties and carbonyls in HFD-induced NAFLD rats. I think it has been reported and accepted that NAFLD is associated with lipid metabolism dysregulation, including fatty acid alteration and lipid oxidation. Therefore, please clearly indicate the original opinion of this work.
The authors did NMR-based lipid profiling. Please show the advantage of this approach compared to MS-based methods, since techniques like LC-HRMS can identify many more metabolites than NMR and give molecular information.
Table 1: It is much more meaningful to focus on the differences within the rats of the same sex.
Figure 2a: Such figures are too difficult to read and do comparison.
Figure 3d: Please explain why between the male groups the SD of CTL was larger than HFD, but between the female groups the SD of HFD was larger than CTL? As the animal number was limited (n = 7 or 8), the sex-dependent/independent findings should be conservative and discreet, they might be coincidence and should be further validated.
Figure 5: Figure legends should be attached near the figures.
Extensive editing of the English language is required.
Author Response
Dear Editor,
Thank you very much for sending us the comments of the reviewers to our manuscript entitled “Lipid Moieties and Carbonyls Are Altered in a Wistar rat Dietary Model of Subclinical Fatty Liver” by Martin-Grau et al. We are glad to see that the reviewers find the manuscript interesting and the results relevant. We thank the reviewers for the thoughtful reading, reviewing, and interest in our work. In the revised version of the manuscript, we have revised the English language, included more data, and extensively modified the manuscript for better clarification of the points raised by the reviewers. We answer their comments in a point-by-point manner in the next pages, with the comment of the reviewer underlined and our answer below.
We think our new version of the manuscript has improved greatly, thanks to the reviewers’ comments. All authors have read and approved the manuscript and no conflicts of interest exist in the study. We hope the manuscript is now suitable for publication in the special issue of Antioxidants. Please, let us know if anything else is needed for quick processing and publication of the manuscript.
Thank you very much for your time, help, and consideration.
Sincerely,
REVIEWER 1
Comment: The authors investigated lipid moieties and carbonyls in HFD-induced NAFLD rats. I think it has been reported and accepted that NAFLD is associated with lipid metabolism dysregulation, including fatty acid alteration and lipid oxidation. Therefore, please clearly indicate the original opinion of this work.
We thank the reviewer for pointing out this important aspect of our manuscript. We agree that in our previous version, the novel aspects of our work were not clearly stated.. . Although lipid profiles in NAFLD have indeed been studied in the past, there are several points that make our study novel and original. This is the first time to our knowledge that:
1) A model is presented in which fat accumulation on the liver is evident, but no liver function biomarkers or other metabolic markers are altered, representing a subclinical/early context of NAFLD
2) Lipid moieties (not lipid species) are analyzed, with a special focus on carbonyls, which are closely related to lipid peroxidation and of interest to the “Antioxidants” reader
3) Lipid peroxidation measurements are normalized to the total content of lipid unsaturation, revealing a potential role of oxidative stress in this context, not visible by MDA alone
4) A full blood lipoprotein profile is measured and differences between sexes are analyzed, providing a unique view of the sex-specific impact of fatty liver on lipid and lipoprotein metabolism
5) Our findings in the preclinical model are supported by the results of our study in a cohort of extremely obese individuals, adding translational value to the study
We truly believe that all these points make our work original, novel, and of interest to the Antioxidants journal reader. Our results suggest a potential novel role of lipid carbonyl content as a biomarker for early detection of NAFLD. We have modified the Abstract and the Discussion sections of our revised manuscript to reflect these novel aspects of our work.
Comment: The authors did NMR-based lipid profiling. Please show the advantage of this approach compared to MS-based methods, since techniques like LC-HRMS can identify many more metabolites than NMR and give molecular information.
We thank the reviewer for bringing this important issue up. Nuclear Magnetic Resonance (NMR) spectroscopy and Mass Spectrometry (MS) spectroscopy are two powerful analytical techniques used currently. It's important to note that NMR and MS are complementary techniques, and the choice between them depends on the specific analytical goals and the nature of the sample being analyzed. Each has its own advantages and is suitable for different types of analysis. Although MS has higher sensitivity and can identify more metabolites, NMR, in all its modalities, has some features that make it the best option for some studies. NMR is truly quantitative and can be applied to liquid, semisolid, and solid samples without any prior sample treatment or destruction which allows to study of the original physicochemical state of species and moieties and facilitates the translation of results to clinical environments. High-resolution magic angle spinning NMR spectroscopy (HR-MAS NMR) is a powerful technique for studying metabolites within different intact tissues, with the same resolution and sensitivity as liquid NMR. Analyzing intact tissues allows the detection of lipids directly in their cellular context, eliminates the need for metabolite extraction, and with it the entailing changes in metabolic composition and properties. HR-MAS used in liver samples reveals accurate information on the composition and properties of lipids and may help us in a greater understanding of lipid metabolism in NAFLD. Finally, the focus of our study is lipid moieties instead of lipid species. Lipid moieties can provide information on the hydrophobic building blocks of lipids, can reveal changes in metabolic clusters that may not be apparent from lipid species, and can help to understand the mechanisms of lipid metabolism and its dysregulation in diseases. In this context, NMR is a structural spectroscopy technique, that provides detailed structural information about organic compounds, including the connectivity of atoms, stereochemistry, and even conformational dynamics, and can help in collecting accurate information about moieties in simple and cheap experiments. We have added some text in the Introduction of the manuscript to further clarify our choice of technique.
Comment: Table 1: It is much more meaningful to focus on the differences within the rats of the same sex.
We agree with the reviewer. It is more important to detect the diet effect than the differences between sexes. In our tables and figures, the differences within rats of the same sex can be easily identified and we emphasize them in our manuscript by listing first the diet effect, within the same sex, in the key to the figures and tables. However, to our understanding and according to our results, the sex effect is also important in this context. The diet effect is rather different between males and females suggesting some protective role present in females that do not appear in males. Therefore, and for the interest of the potential reader of the article, we also show the statistical significance of sex in our figures and tables.
Comment: Figure 2a: Such figures are too difficult to read and do comparison.
We understand the concern of the reviewer. Such figures could be intimidating at first. However, the reading of Figure 2a is very similar the same as the reading of Figure 2c. Figure 2a contains the mean differences between control and diet animals with lines showing the 95% confidence intervals whereas Figure 2c contains the mean value per group with lines showing the 95% confidence intervals. The apparent complexity can come from the circle shape. However, it is impossible to fit the same information in a format like Figure 2c without taking several pages of the manuscript and having a very large and difficult-to-read figure. In addition, having a circus plot for each sex, helps in detecting the differences within the same sex, as important as the reviewer pointed out in the previous comment, and helps in comparing the pattern of differences between sexes. We have added some text to the figure legend of the revised version of the manuscript for clarity.
Comment: Figure 3d: Please explain why between the male groups the SD of CTL was larger than HFD, but between the female groups the SD of HFD was larger than CTL? As the animal number was limited (n = 7 or 8), the sex-dependent/independent findings should be conservative and discreet, they might be coincidence and should be further validated.
We thank the reviewer for pointing out this issue. Although standard deviation may represent the dispersion in the values of a group, it could also be affected by outliers and the number of samples, as the reviewer suggests. The animal number cannot be modified because it was recommended by our statistician after sample size calculations and the number was approved by our Ethical Committee for Animal experimentation. We agree that differences in standard deviation between groups could reflect unstable biological situations or just be a statistics artifact. We do not believe that our sex-dependent/independent results could be a coincidence because of the statistical significance detected, the consistency of the different findings, and the support found in humans. Nevertheless, and according to the reviewer's suggestion, we have modified the manuscript to be more conservative and less assertive.
Comment: Figure 5: Figure legends should be attached near the figures
We are grateful to the reviewer for pointing out this formatting error. It has been corrected in the revised version of the manuscript
Reviewer 2 Report
1. In my opinion, the title and abstract should better reflect the study performed. Human cohort study should be mentioned in some way (not only in the last sentence of the abstract without mentioning it in the material and method part of the abstract). However, I do not understand why you put the results of two studies in one article. In my opinion, this human cohort study should be describe in details as a new article, because the results are really interesting.
2. 2.2. Human cohort of extreme obesity - this chapter should be rewritten, especially the part concerning blood sampling, because now it is not clear written / English requires correction. You should explained how the presence/absence of insulin resistance was determined in the patients. There is important parameters found in the table S2, but in the chapter, the methods are not provided (eg. glucose measurement, the equation for HOMA-IR, etc.). I advice to put this table in the main text, not in the supplementary. Generally, the entire chapter Material and methods should be corrected to be more readable.
3. Line 220 and others: p should be lower than 0.05 (a dot, not a comma should be used in numbers).
4. All abbreviations used in tables/figures should be explained in the footnotes/legends independently on the main text.
5. It would be better to use body/live mass, not weight in the whole manuscript. And remember to correct all numbers - not comas, but dots.
6. English should be corrected in the whole manuscript.
7. Line 442 - you mentioned that insulin resistance was not present in rats, these results/ values are not shown in the Results section.
8. Limitations of the study should be added to Discussion.
English is not very poor, but requires moderate correction by a professional.
Author Response
Dear Editor,
Thank you very much for sending us the comments of the reviewers to our manuscript entitled “Lipid Moieties and Carbonyls Are Altered in a Wistar rat Dietary Model of Subclinical Fatty Liver” by Martin-Grau et al. We are glad to see that the reviewers find the manuscript interesting and the results relevant. We thank the reviewers for the thoughtful reading, reviewing, and interest in our work. In the revised version of the manuscript, we have revised the English language, included more data, and extensively modified the manuscript for better clarification of the points raised by the reviewers. We answer their comments in a point-by-point manner in the next pages, with the comment of the reviewer underlined and our answer below.
We think our new version of the manuscript has improved greatly, thanks to the reviewers’ comments. All authors have read and approved the manuscript and no conflicts of interest exist in the study. We hope the manuscript is now suitable for publication in the special issue of Antioxidants. Please, let us know if anything else is needed for quick processing and publication of the manuscript.
Thank you very much for your time, help, and consideration.
Sincerely,
Comment: In my opinion, the title and abstract should better reflect the study performed. Human cohort study should be mentioned in some way (not only in the last sentence of the abstract without mentioning it in the material and method part of the abstract). However, I do not understand why you put the results of two studies in one article. In my opinion, this human cohort study should be describe in details as a new article, because the results are really interesting.
We are very grateful to the reviewer for the interest in our study and the positive words about it. Although it is true that we presented two studies in one single manuscript, we think that human studies are important to complement and support findings in experimental models, especially in the field of lipid research. First, experimental models may not fully reflect the complexity and diversity of human physiology and pathology. Second, animal models can generate hypotheses and mechanistic insights, but they need to be confirmed and tested in human subjects before they can be applied to the diagnosis, prevention, or treatment of diseases. Finally, animal models often have controlled and standardized conditions, which may not capture the variability and heterogeneity of human populations. We thought it could be appropriate to add the human section to give greater value to the article and provide stronger bases for subsequent more complete human studies.
We agree with the reviewer about the completeness of the title and abstract and we have modified both trying to better capture the extent and value of our study.
Comment: 2. 2.2. Human cohort of extreme obesity - this chapter should be rewritten, especially the part concerning blood sampling, because now it is not clear written / English requires correction. You should explain how the presence/absence of insulin resistance was determined in the patients. There is important parameters found in the table S2, but in the chapter, the methods are not provided (eg. glucose measurement, the equation for HOMA-IR, etc.). I advice to put this table in the main text, not in the supplementary. Generally, the entire chapter Material and methods should be corrected to be more readable.
We thank the reviewer for the suggestion to make the manuscript more readable. It is true that some sections from Material and Methods were not explained in full detail and only a reference to a prior study was included. We have added more information in each paragraph of the Material and Methods for clarity. As suggested by the reviewer, we have paid special attention and rewritten the human cohort paragraph (adding anthropometrical values, biochemical measurement, how the presence/absence of insulin resistance was determined, the equation for HOMA-IR, etc.). We have also inserted Table S2 of the original supplementary material as Table 3 in the main text of the revised manuscript.
Comment: 3. Line 220 and others: p should be lower than 0.05 (a dot, not a comma should be used in numbers).
We thank the reviewer for pointing out this general mtypo. It has been corrected in the revised version of the manuscript.
Comment: 4. All abbreviations used in tables/figures should be explained in the footnotes/legends independently on the main text.
We thank the reviewer for this comment which improves clarity. WE left out some abbreviations in the original manuscript but all of them have been revised and included in the revised version of the manuscript.
Comment: 5. It would be better to use body/live mass, not weight in the whole manuscript. And remember to correct all numbers - not comas, but dots.
We thank the reviewer for this suggestion, which better fulfils international agreements and scientific thinking. We have change weight for mass in all the revised manuscript.
Comment: English should be corrected in the whole manuscript.
We thank the reviewer for this recommendation. We hired a Scientific English Editor to review the whole manuscript and we think that the English has been vastly improved
Comment: Line 442 - you mentioned that insulin resistance was not present in rats, these results/ values are not shown in the Results section.
We thank the reviewer for noticing and alerting about this. We have added the values in Table 1, mentioned them in Material and Methods, explained them in Results and commented them in Discussion sections. HOMA-IR is difficult to be calculated and applied in rats because there are no common criteria as in humans. That’s why we have added only the insulin values in Table 1 and not HOMA-IR values.
Comment: Limitations of the study should be added to Discussion.
We thank the reviewer for this suggestion which improves the quality of virtually any manuscript published in a scientific journal. We have added the main limitations to this study in the last paragraph of the Discussion section.
Round 2
Reviewer 1 Report
The authors did extensive modifications to improve the paper quality. Yet, the figures are not clear enough, especially figs 2, 3, 4.
English has been improved. Now much better.
Author Response
We are very grateful to the reviewer for the appreciation of our efforts to improve the manuscript. However, we do not fully understand if the comment about the clarity of Figures 2,3 and 4 refers to the image quality, the figure content, or the figure legend. We have improved these three aspects of figures 2, 3, and 4. New versions of figures 2, 3, and 4 have higher font size and detail for better reading and understanding and higher resolution for quality. We have also expanded the figure legends with a larger explanation of the content of the figures for better readability. We hope that the revised version is much clearer.
Reviewer 2 Report
Dear Authors,
Thank you for your effort and significant revision of your manuscript. You have followed all my comments and explained issues I asked about. I think now the article can be published in this version.
English has been improved significantly. Some minor correction (typos etc.) could be provided, but I think it could be in the proofs mode.
Author Response
We thank the reviewer for the careful reading and the positive comments.